# Nitrogen and phosphorus fertilization consistently favor pathogenic over mutualistic fungi in grassland soils

Ylva Lekberg [1,2,10 ✉], Carlos A. Arnillas [3,10], Elizabeth T. Borer [4], Lorinda S. Bullington [1], Noah Fierer[5,6], Peter G. Kennedy[7], Jonathan W. Leff [8], Angela D. Luis[2], Eric W. Seabloom [4] & Jeremiah A. Henning[4,9]

Ecosystems across the globe receive elevated inputs of nutrients, but the consequences of this for soil fungal guilds that mediate key ecosystem functions remain unclear. We find that nitrogen and phosphorus addition to 25 grasslands distributed across four continents promotes the relative abundance of fungal pathogens, suppresses mutualists, but does not affect saprotrophs. Structural equation models suggest that responses are often indirect and primarily mediated by nutrient-induced shifts in plant communities. Nutrient addition also reduces co-occurrences within and among fungal guilds, which could have important consequences for belowground interactions. Focusing only on plots that received no nutrient addition, soil properties influence pathogen abundance globally, whereas plant community characteristics influence mutualists, and climate influence saprotrophs. We show consistent, guild-level responses that enhance our ability to predict shifts in soil function related to anthropogenic eutrophication, which can have longer-term consequences for plant communities.

[1] MPG Ranch, Missoula, MT, USA. [2] Department of Ecosystem and Conservation Sciences, University of Montana, Missoula, MT, USA. [3] Department of Physical and Environmental Sciences, University of Toronto – Scarborough, Scarborough, Canada. [4] Department of Ecology, Evolution, and Behavior, University of Minnesota, St Paul, MN, USA. [5] Department of Ecology and Evolutionary Biology, University of Colorado, Boulder, CO, USA. [6] Cooperative Institute for Research in Environmental Sciences, University of Colorado, Boulder, CO, USA. [7] Departments of Plant Biology and Ecology, University of Minnesota, St Paul, MN, USA. [8] Independent Researcher, Boulder, CO, USA. [9] Department of Biology, University of South Alabama, Mobile, AL, USA. [10]These authors contributed equally: Ylva Lekberg, Carlos A. Arnillas. ✉email: ylekberg@mpgranch.com

Ecosystems across the globe are receiving elevated inputs of nitrogen (N) and phosphorus (P) from agriculture and urban activities[1], and atmospheric N deposition has increased threefold from pre-industrial levels and will likely increase in the foreseeable future[2]. Many plant communities are N limited[3], and additional N can promote plant productivity if P is non-limiting[4,5]. However, growth responses to nutrient addition are not always positive[6] and can decline over time[7]. This is often attributed to altered plant community composition, involving loss of dominant species and an increase in ruderal plants[7], but there is growing recognition that nutrient addition may also alter the composition and function of soil microbial communities, which may feedback to limit plant community productivity[8,9].

As mutualists, pathogens, and saprotrophs, soil fungi regulate key ecosystem processes, including plant primary productivity and carbon mineralization and sequestration[10,11]. Global patterns in fungal guild abundance are related to climate[12], but guilds are also affected by soil fertility[11]. For example, most herbaceous plants, shrubs, and trees in temperate and tropical habitats form a root symbiosis with the putative mutualists, arbuscular mycorrhizal fungi (AMF)[13]. Inside roots, AMF exchange nutrients (especially P) acquired in soil for plant carbon (C)[13]. Where P is plentiful, AMF abundance is often lower due to reduced C allocation from plants[13]. The addition of N can benefit AMF if it exacerbates plant P limitation[14], but may be suppressive if nitrophilic, ruderal plants replace plants that allocate more C to AMF[6,7]. Thus, responses by AMF likely depend on the extent to which nutrient addition alleviates plant deficiencies and alter plant communities. Like AMF, interactions between plants and fungal pathogens are complicated and depend on both host and pathogen responses to environmental conditions[15]. Most research to date has been conducted on cultivated plants, and less is known about the role of pathogens in natural communities, although recent work highlights their importance for maintaining plant diversity[16]. One emerging pattern from work in natural systems is that pathogens often thrive in resource-rich environments[9,17], and fertilizer addition has been linked to increased disease incidence in plants[18,19]. This effect is likely greater with N than P, because P addition can enhance plant vigor, which tends to decrease susceptibility to pathogens[18,19]. Fungal saprotrophs have received more attention recently because of their key role in soil C flux and storage[20–22] where relatively small changes in abundance and activity could have outsized consequences for C budgets. While specific responses to P addition are poorly understood, N addition can affect saprotrophic activity and decomposition rates. However, such effects are hard to predict, can change over time, and depend on the saprotrophic community composition, litter quality, soil fertility, and N supply rate[20,23–26]. For example, N addition may accelerate initial decomposition rates but retard turnover of more recalcitrant litter due to shifts in oxidative enzymes[23,26]. As such, it has been suggested that anthropogenic N deposition can promote C sequestration[5,25].

Most previous studies have focused on individual guild responses at single locations or have spanned multiple vegetation types, which complicates direct comparisons among locations within a single vegetation type. To better predict current and future functioning of soil microbial communities, we need to understand how altered nutrient availability influences fungal guilds, if responses are consistent across locations, and the nature of the underlying drivers.

In this work, we combined a previously published dataset[27] that documented phylum-level responses (Ascomycota, Basidiomycota, Glomeromycota, and Zygomycota) to N and P addition with novel data on AMF colonization in roots collected from the same soil cores. Because only Glomeromycota can be assigned into a functional guild at the phylum level of resolution, these higher taxonomic groups offer limited information about possible functional consequences of nutrient addition. We therefore used taxonomic information more closely related to species, and assigned potential function to taxa using FUNGuild[28]. This allowed us to quantify responses by AMF, putative pathogens, and saprotrophs after 1–4 years of experimental N and P addition at 25 Nutrient Network (NutNet)[29] grassland sites across four continents (Supplementary Fig. 1). We predicted that AMF would be suppressed by P and respond less consistently to N, whereas pathogens would be promoted by N and possibly be suppressed by P. Given the context-dependent nature of saprotroph responses in the literature, we expected less consistent responses by saprotrophic fungi. To identify the underlying mechanisms, we used structural equation modeling (SEM) to determine if guild responses were directly driven by N and P (while accounting for initial soil conditions), or indirectly mediated by plant community responses. We predicted that responses would primarily be driven by shifts in plant communities given the dependency on plants by all three guilds. The novel data on AMF colonization in roots allowed us to determine if N or P addition shifted AMF biomass allocation between roots and soil. Finally, we used co-occurrence networks[30] to assess how nutrient addition may change potential interactions within and among guilds. To explore if differences in guild abundance across sites were related to inherent differences in soil properties, plant communities, and climate, we used a mediation test[31] restricted to unfertilized control plots.

We find that nutrient addition promotes pathogens, suppresses AMF, and reduces co-occurrences within and among fungal guilds in ways that appear primarily mediated by shifts in plant communities. We also document global distribution patterns of fungal guilds, with soil properties influencing pathogens, plant community characteristics influencing mutualists, and climate affecting saprotrophs. Our results demonstrate that fungal guilds respond predictably to nutrient addition despite substantial differences in plant and fungal communities, climatic conditions, and edaphic properties across grasslands. The generality of these patterns contributes to a growing body of knowledge that may ultimately help us build better models to identify areas where disease prevalence or severity could be particularly high, and where soils are more likely to be carbon sources than sinks under anthropogenically altered environmental conditions.

## Results and discussion

**Guild responses to N and P additions.** Independent and combined N and P addition promoted fungal pathogens as we observed an average increase of 140% in N + P plots relative to control plots in relative pathogen abundances. The effect of N aligns with previous work in mostly cultivated systems where N addition has increased disease severity[18,19]. The independent promotion of pathogens by P was surprising given that P has more often had no effect[18] or has been associated with improved plant health and reduced disease[19]. Future work should assess if this is largely driven by a subset of taxa, as pathogens differ in their response to fertilizer[18]. Unfortunately, this could not be done here due to an insufficient number of sequences for robust community analyses.

We predicted that P would suppress AMF, but this only occurred when P was added together with N (−33% relative to control plots), whereas independent addition of N or P had no effect (Fig. 1). Because AMF rely on plants for all their C, this general suppression of AMF with N and P indicates that plants can somewhat adjust C flow to AMF and reduce the risk of parasitism[32] when nutrient availability is high. Direct links

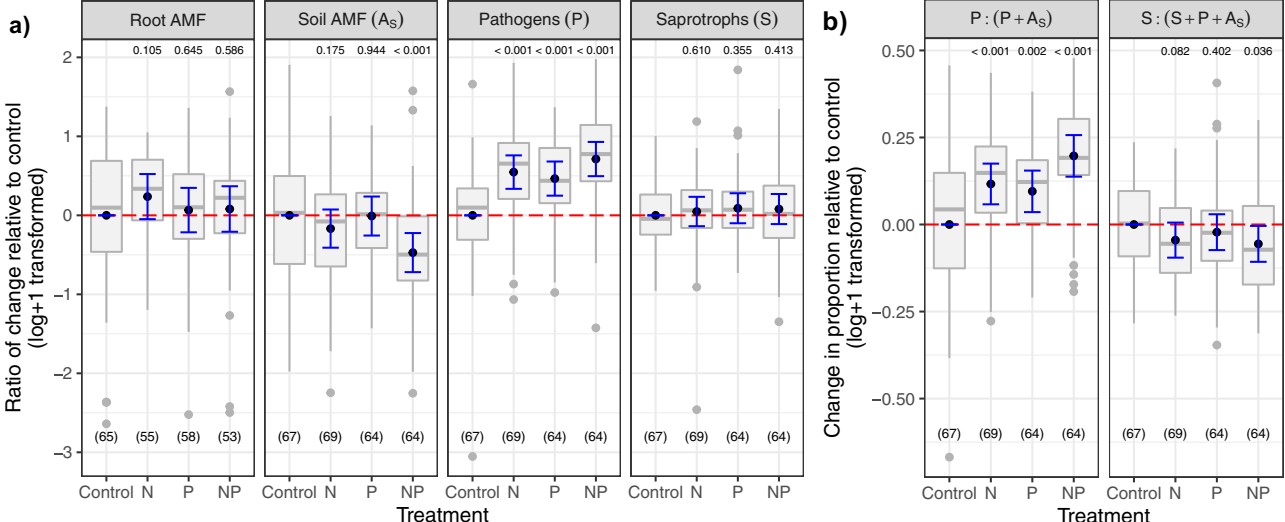

**Fig. 1 Responses by fungal guilds to nitrogen (N), phosphorus (P), and N + P addition across grasslands. a** Colonization by AMF in roots based on microscopy (Root AMF), and soil abundance of AMF (Soil AMF, $A_S$, putative pathogens (P), and saprotrophs (S) based on sequence numbers in comparison to the control plots. **b** The P:(P + $A_S$) quantifies shifts in pathogen to mutualist ratios, and the S:(S + P + $A_S$) highlights shifts in the producer and decomposer food webs[61] and how nutrient addition changes the relative abundance of fungi associated with living and dead plants. The black dot indicates the estimated mean effect, the blue bars the 0.95 confidence interval of the mean, while the gray bar and dots represent the boxplot and corresponding outliers of the partial residuals. Boxplots denote median value (black bar), with hinges (gray) representing the 25th and 75th percentiles, and lines extending to the 1.5× interquartile range. Numbers at the bottom indicate the number of observations included in each treatment. Numbers at the top indicate the *p*-value based on the two-sided t-statistic of the estimated treatment mean effect compared to the control plots.

between fungal guilds and plant–soil feedbacks have been shown previously[11]. The promotion of pathogens and—to a lesser extent —the suppression of AMF with nutrient addition observed here may help explain why plant biomass responses to fertilizer decline with time[4,7]. It also supports predictions that the effects of soil biota on plant growth are more negative in resource-rich environments[9,11,33].

To acquire C, AMF must colonize roots, but relative biomass allocation between roots and soil differs among fungal taxa and environmental conditions[13,34]. AMF occupancy of both roots and soil differed across sites here, but unlike AMF soil colonization, root colonization was not suppressed by combined N and P addition. On the contrary, root colonization appeared to have been slightly promoted by N ($P = 0.105$, Fig. 1), which suggests a shift in fungal allocation with N addition where a greater proportion of total fungal biomass was inside, not outside, the root. Nitrogen addition can alter AMF communities from taxa that grow extensively in soils to those that preferentially occupy roots[34–36], but see ref. [37]. Due to the generally low AMF sequence counts, we could not assess if shifts in composition accompanied the slight shift in allocation, here. Regardless, AMF that allocate biomass preferentially inside rather than outside roots are sometimes considered less beneficial with a lower capacity to acquire P[35,38], but they can provide protection against pathogens[39,40]. Thus, we speculate that an intriguing alternative hypothesis to nutrient-induced parasitism is that N promotes pathogens, which initiates a switch in AMF function from resource acquisition to pathogen protection. Testing the pathogen protective ability of AMF that are promoted by N in soils where P is non-limiting would be a fruitful area for future research, because, if true, it would alter how we view the mutualism-to-parasitism continuum of this symbiosis[41].

Compared with pathogens and mutualists, nutrient addition did not alter the relative abundance of fungal saprotrophs (Fig. 1 and Supplementary Fig. 1), concordant with an earlier study conducted along a soil fertility gradient[42]. This lack of response was evident despite an increase in productivity and may be

because litter biomass, and thus substrate inputs, did not increase with N or P addition (Supplementary Table 1). This lack of response is also in agreement with the observation that soil C concentration has not changed in response to nutrient addition in many of the same grasslands[22]. It is possible that the context-dependency of responses to N addition outlined earlier may limit directional responses due to differences in climate[22], or that measurable responses may simply be slow to manifest. However, responses across all three guilds did not depend on the number of years elapsed since the start of the nutrient addition ($P > 0.05$), suggesting that factors other than time are stronger determinants of fungal responses to nutrient addition. Nonetheless, some of sites in the Nutrient Network have been maintained for >10 years, and repeated sampling could assess whether differences in the first 1–4 years decrease over time as communities acclimate to the novel nutrient conditions, stabilize and remain the same, or become more pronounced.

**Direct or indirect effects of N and P on fungal guilds**. In accordance with our predictions and previous work[14,19], the SEM supported plant community mediation of fungal guild responses in most cases. AMF were suppressed by an increased plant biomass, saprotrophs were promoted by greater root biomass, and both AMF and pathogens responded to shifts in community dissimilarity. Here AMF abundance was suppressed as the phylogenetic distance increased within plant communities, whereas pathogen abundance was promoted when plant communities became more dissimilar from control communities (Fig. 2). How plant communities may drive the observed shifts is unknown. But if nutrient addition changes plant communities toward taxa with more ruderal traits, this can promote pathogens and suppress mutualists[11], because ruderal plants allocate less C to AMF and are more susceptible to disease[19,43–45]. Two previous studies that focused on many of the same sites show that N and P addition reduces root:shoot ratios, promotes nutrient concentrations in leaves, and increases specific leaf areas[46,47]; traits that are all

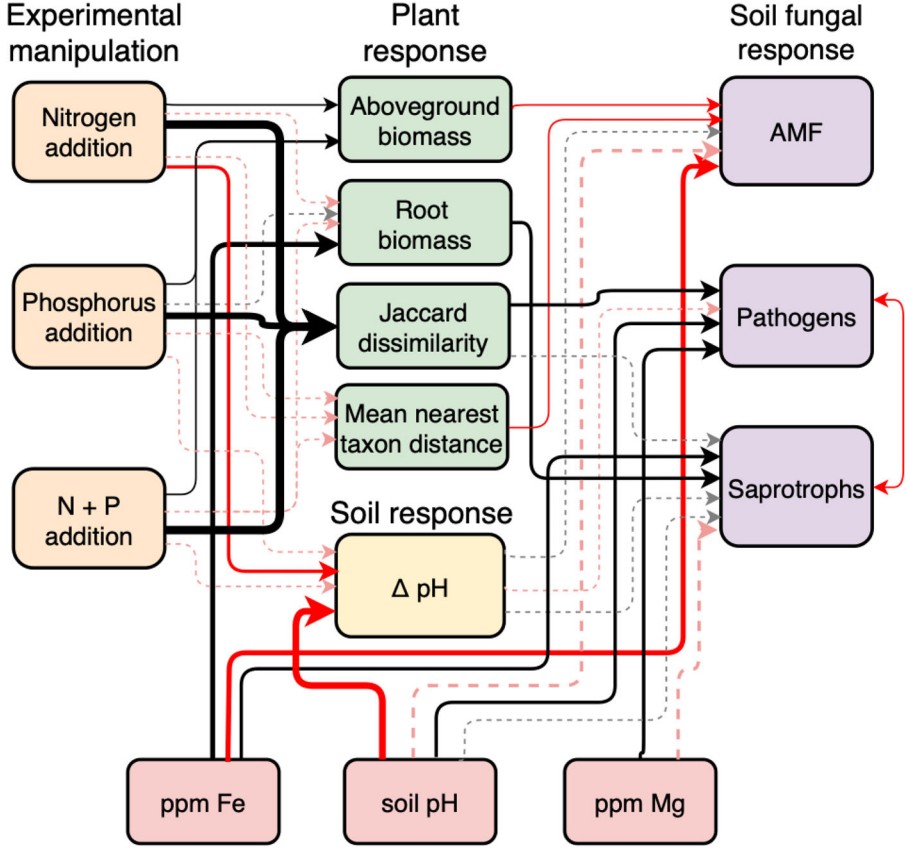

**Fig. 2 SEM assessing direct responses by guilds or indirect effects mediated by plant communities and pre-treatment soil properties.** Black lines indicate significant ($P < 0.05$) positive relationships and red lines negative relationships and the strength of these relationships is indicated by the width of the arrows where dotted lines indicate pathways that are non-significant ($P > 0.05$) but improved the model-fit (Supplementary Table 2). Because some predictor or response variables were missing for some sites, the SEM analyses were restricted to 15 sites with complete data matrices.

associated with faster life histories[48]. Given that plant richness and functional groups are often highlighted as potential drivers of plant-soil feedbacks[49], it is noteworthy that plant functional groups played only a minor role compared to overall community descriptors here, and that the effect of plant richness was mostly mediated by changes in phylogenetic distance within communities (MNTD). Whether shifts in fungal guilds precede—and possibly cause—shifts in plant communities, or if fungal guilds respond to altered plant species and traits will require detailed analyses of plant and fungal community changes over time. We also cannot rule out a direct effect of N and P on pathogens, because a model replacing plant community similarity with the direct effect of treatment (while keeping soil properties) performed similarly (Supplementary Table 2). Careful experimentation could disentangle the potential bidirectional interactions of plant community and soil chemical properties on fungal guild abundances, especially pathogens.

One unexpected finding was the relatively strong effect of pre-treatment soil nutrient concentrations (Fig. 2 and Supplementary Table 3). For instance, iron was associated with a higher saprotroph abundance and root biomass but a lower AMF soil abundance and root colonization (Supplementary Fig. 2). Even though we found directional responses to N and P addition by putative pathogens and AMF across sites, among-site variability and pre-treatment soil conditions explained more variability than the N and P effects or the differences among plant communities at the time of sampling. Several soil nutrients were correlated

(Supplementary Figs. 3 and 4), so the specific effect of individual nutrients is difficult to determine. Further, pre-treatment soil conditions are likely correlated with initial plant communities, so we cannot isolate plant effects from soil effects on the fungal guilds with the available information. It is safe to say, however, that soil chemistry is an important mediator of fungal guild abundances. Soil pH is a known driver of fungal communities[50,51], and pre-treatment pH influenced pathogens but had little effect on saprotrophs and AMF (Fig. 2 and Supplementary Fig. 2). Why pH —or other soil properties that correlate with soil pH—affects one guild more than another, and in different directions, is unknown but could have consequences for identifying sites where disease prevalence or severity could be particularly high.

We found more complex networks in control plots than plots receiving nutrients (Fig. 3 and Supplementary Table 4 for identities). Network complexity can be reduced by perturbation[52,53] and fungal communities may respond to fertilization as a disturbance (possibly short-term), where the enriched soil nutrient environment is completely altered and homogenized in ways that could alter the available niches. Our results could be a product of habitat filtering where taxa respond similarly to environmental shifts, but do not necessarily interact[54]. As such, more research is needed to disentangle the cause and consequences of these changing patterns. If nutrient addition reduces interspecific interactions, it could have functional consequences, as more connected soil networks are more efficient at C uptake and nutrient cycling[52].

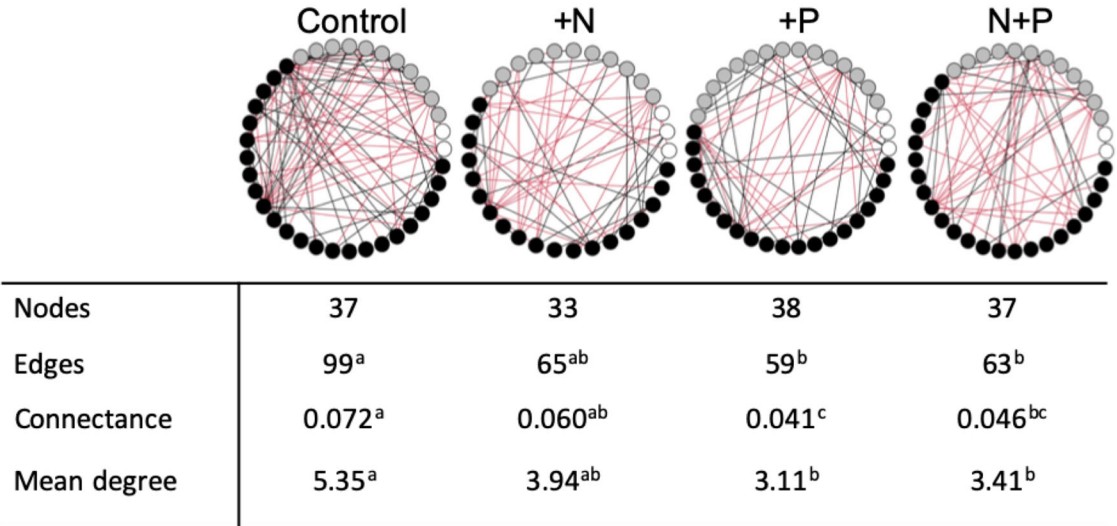

| | Control | +N | +P | N+P |
|---|---|---|---|---|
| Nodes | 37 | 33 | 38 | 37 |
| Edges | 99[a] | 65[ab] | 59[b] | 63[b] |
| Connectance | 0.072[a] | 0.060[ab] | 0.041[c] | 0.046[bc] |
| Mean degree | 5.35[a] | 3.94[ab] | 3.11[b] | 3.41[b] |

**Fig. 3 Co-occurrence network analyses on fungal genera belonging to mutualists (white circles), pathogens (gray circles), and saprotrophs (black circles).** The number of nodes is the number of genera that had significant correlations with other genera, edges represent the number of significant correlations (positive indicated by black and negative indicated by red), connectance is the proportion of all possible edges that are present in each network, calculated as (number of edges)/(number of nodes)[2], and mean degree is the average number of edges per node. Different superscript letters indicate statistically significant differences. All 25 sites were included in the co-occurrence analysis.

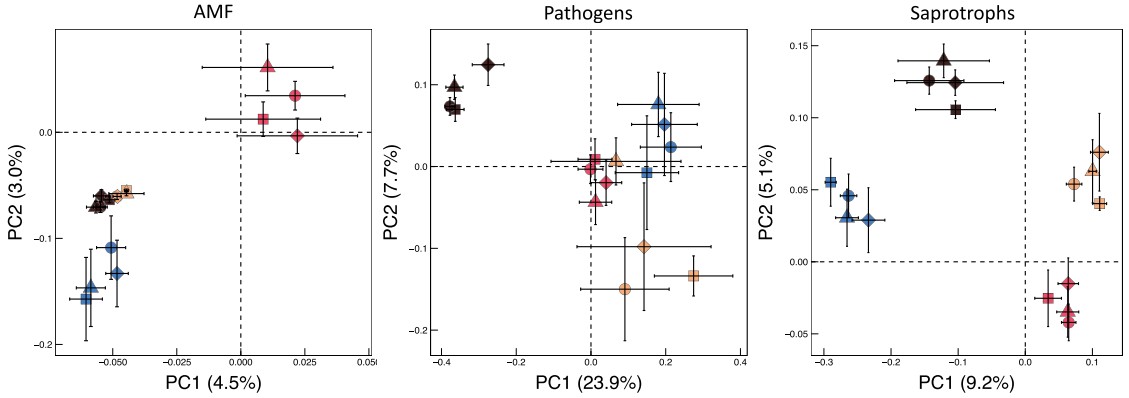

**Fig. 4 Principal coordinate analyses of Bray–Curtis distances of Hellinger transformed relative abundance data showing dissimilarities among communities for mutualists (AMF), pathogens, and saprotrophs occurring in North America (pink, $n = 16$), Europe (blue, $n = 3$), Africa (maroon, $n = 3$), and Australia (peach, $n = 3$) in either control plots (circles) or plots that had received nitrogen (squares), phosphorus (triangles), or both (diamonds).** Error bars around the means represent standard error. Communities of all guilds differed among continents using PERMANOVA ($p < 0.001$), but nutrient treatments had no consistent effect on composition, most likely due to high community turnover among sites and low sequence numbers resulting in low power. Results were not different when dissimilarities were assessed based on Raup-Crick transformed presence/absence data.

**Global drivers of fungal guilds.** We observed clear biogeographical patterns of fungal taxa within all guilds (Fig. 4), suggesting that responses to N and P addition occurred despite substantial differences in taxonomic composition. Also, sites differed in their relative guild abundances (Supplementary Table 5). Some sites contained many saprotrophs, but very few AMF, and vice versa. When restricting analyses only to control plots (those receiving no fertilizer inputs) to assess potential underlying drivers of these distribution patterns, we found that each guild followed a distinctive pattern (Table 1). Similar to the SEM, plant communities were the most important predictor of AMF (root and soil colonization), whereas pathogens were more affected by soil properties (Supplementary Table 6 and Supplementary Fig. 2). In contrast, climate was the most important driver of saprotroph abundance. It is also possible that dissimilarities in plant community composition could be important for predicting differences in pathogen abundances, similar to our SEM and earlier findings targeting the whole fungal community[55]. Unfortunately, we could not include a measure of plant community dissimilarity in these analyses as, per definition, it relies on pairwise comparisons and a "reference" community, analogous to the control plots used to calculate the treatment effect in the SEM. Regardless, our results build on previous work across biomes showing differential responses among fungal guilds to plant community characteristics, soil edaphic properties, and climate[50,56]. De-aggregating bulk fungal responses and examining differences among fungal guilds will help us better predict soil function and responses to future perturbations.

**Limitations.** While we show clear directional effects of N and P addition on fungal guilds, these findings are based on relatively short-term responses and rely on sequence counts and their link to potential function. This approach assumes that sequence counts accurately reflect abundance or biomass and that potential function informs actual function, which may not always be the

**Table 1 Total variability ($R^2$) explained from each fungal guild by the most parsimonious predictive model and amount of variability explained by predictor type.**

| Fungal guild | N | Total $R^2$ | Location | Climate | Soil | Biomass (BM) | Biodiversity (BD) | Plants (BM & BD) |
|---|---|---|---|---|---|---|---|---|
| *AMF* | | | | | | | | |
| Root | 20 | 0.608 | – | – | 0.147 | 0.390 | – | **0.390** |
| Soil | 19 | 0.811 | 0.302 | – | – | 0.443 | 0.195 | **0.614** |
| Pathogens | 19 | 0.499 | – | – | **0.499** | – | – | – |
| Saprotrophs | 19 | 0.740 | – | **0.509** | 0.149 | – | – | – |

The amount explained by predictor type was estimated as the difference between the $R^2$ of the full model minus the $R^2$ of the model without those variables. Plant community effect was estimated independently dropping biomass (BM) and biodiversity (BD) variables, and also dropping biomass and biodiversity variables together (BM & BD). In bold, the largest fraction for each fungal guild.

case[57,58]. For example, AMF can sometimes be parasitic when costs outweigh benefits[41], and pathogens can occur in host tissues without causing disease[59]. We argue that these potential discrepancies, including potential PCR biases[60], apply equally to all treatments and in comparisons across sites, and that our method can be used to assess relative shifts across treatments and global distributions. The use of relative sequence counts is another potential issue where responses by one guild may constrain responses by another given the proportional nature of these data. While the SEM indicated that some guilds are correlated (Fig. 2), in accordance with previous surveys[28], ~40% of the fungal sequence data could not be functionally assigned, so responses among the guilds we tested should therefore be relatively decoupled. Interactions between the producer and decomposer food webs also are expected[61], and correlations among fungal guilds have been documented previously[62], suggesting that the correlations observed here represent actual ecological responses. Most importantly, the shifts in pathogen to mutualist ratios with nutrient addition we observed is consistent with previously documented changes in plant responses to soil biota[9,11]. For example, plants inoculated with soil from fertilized fields were 50% smaller than plants grown in soil collected from unfertilized fields (estimated from Revillini et al.[9], Fig. 4). Two of the four sites in Revillini et al. (Konza LTER and Short Grass Steppe LTER) showed a more than tenfold increase in pathogen to mutualist ratios between control and N + P addition plots in our analyses. Thus, while we cannot unequivocally support a direct link between pathogen abundance and disease development or growth suppression here, this has been shown previously[63,64], supporting the likely ecological relevance of the results presented here.

## Methods

**General background and FUNGuild analyses**. We used a published fungal dataset generated from 25 NutNet grassland sites[27] that are distributed world-wide and where researchers follow the same treatment and sampling protocols. In brief, each grassland site is situated in a relatively homogeneous ~1000 m² area divided into three blocks. Each block is made up of 5 ×5 m plots that are surveyed annually for plant community composition and productivity, and a subset of plots receive applications of 10 g of either N [$(NH_2)_2CO$] or P [$Ca(H_2PO_4)_2$] or both m$^{-2}$ per year. Soil samples for fungal community analyses were collected from control plots and plots with N and P added alone and in combination in 2011 or 2012, 1–4 years after the initial nutrient addition. DNA was extracted and amplified using fungal-specific primers (ITS1F/ITS2) targeting the internal transcribed spacer (ITS1) region. For more details on experimental design as well as soil sampling, DNA extraction, amplification, and bioinformatics see Borer et al.[29], Leff et al.[27], Prober et al.[55], and supplemental material.

We matched taxonomic identities of fungal sequences randomly rarefied to 485 sequences per plot with ecological guilds using the expert-curated database FUNGuild[28]. This rarefaction level represents a trade-off between keeping as many sites as possible in our analyses while characterizing most taxa within each treatment at each site. This rarefaction was also chosen as it was the same as in Leff et al.[27], thus allowing direct comparisons with that publication. While this rarefaction level did not result in an exhaustive characterization of all taxa at all sites, it did capture abundant taxa in all sites (Supplementary Fig. 6). We then assessed if sequence numbers belonging to different guilds varied across sites and

among control plots and those receiving N, P, or N + P. Of the 164,900 total sequence reads, 60% belonged to taxa annotated with a guild assignment. These assignments were further subset into three guilds: highly probable and probable arbuscular mycorrhizal (AMF, putative mutualists), plant pathogenic fungi, and saprotrophic fungi. For the latter, both the soil saprotroph and undefined saprotroph guilds were included. The specific substrate or habitat of many saprotrophic fungi is undefined, and by only including soil and undefined saprotrophs, we sought to exclude those taxa known to associate with non-representative substrates such as wood or dung. Total sequence counts were 5659 for AMF, 13,263 for plant pathogens, and 24,333 for saprotrophs. To complement sequence data from soils, we also quantified the abundance of AMF in roots using trypan blue and the gridline intersect method[13]. For this additional analysis, we included four additional sites that contained roots but not sequence data ($n = 29$).

**Responses by fungal guilds and structural equation model**. To understand overall trends in fungal communities in response to nutrient addition, we compared fungal guild sequences across nutrient addition treatments using a linear mixed-effects model and incorporating site and block within sites as random effects using *lmerTest* package. Fungal guild sequences were log +1 transformed to meet assumptions of normality. We calculated the partial effect of the treatment and the partial residuals using the *visreg* package (Fig. 1). To assess the potential mechanisms driving the treatment effect, we included three sets of covariates to determine if any of them could drive the observed patterns: post-treatment vegetation, pH and root biomass, and pre-treatment soil conditions (Supplementary Table 5). For each guild, we eliminated one of each pair of highly correlated variables identified using a variance inflation factor (VIF) and, to obtain a more parsimonious model, we retained only significant variables. Post-treatment vegetation was described using functional group descriptors or whole community descriptors, and the best model in each case was compared between them. For every regression, we required at least two control plots and eight total samples per site for inclusion. Not enough samples had all the information needed so we dropped the random block factor and excluded root colonization in the model. Also, because some predictor or response variables were missing for some sites, the SEM analyses were restricted to 15 sites with complete data matrices.

The significant predictors ($p < 0.05$) from each of the three groups of covariates were pulled together to build the final regression model for each fungal guild. We combined the three regressions obtained for AMF, pathogens and saprotrophs into a structural equation model (SEM), including the effect of treatment and each significant predictor ($p < 0.05$) using the *piecewiseSEM* package[65]. We modified these final models and obtained the marginal and conditional $R^2$ to test if adding treatment effects or replacing the plant community information with treatments improved the models. Finally, to assess the overall explanatory power of each of those variables, we estimated the marginal and conditional $R^2$ of models including plant community and pre-treatment soil conditions only.

**Global mediation test**. Because the relative abundance within each guild differed among sites (Supplementary Table 5), we ran a mediation test to assess whether differences in soil (including total N range: 0.05–1.6% and available P range: 7–248 mg kg$^{-1}$ soil), climate, or plant communities better predict the observed abundance trends. We used control plots only and averaged observed values across years (years 0–4 after establishment where available) to reduce the effects of plot-to-plot and year-to-year variation. Thus, each site was represented by a single value for each predictor and response. We identified the best predictors for each guild from each set of covariates (plant communities, soil, climate, location), selecting variables using a stepwise backward selection model, using BIC to identify the most parsimonious model. Then, for each combination of fungal guild and predictor set, we tested if the best predictors of another set could improve the model (provide new information) or decrease the significance of some of the original predictors (which could suggest that one of the variables added mediates the effect of the original predictor). See Supplementary Information for detailed methods.

**Co-occurrence network analyses**. To examine co-occurrence patterns, first, we grouped all taxa that were identified to guild by genus and looked for significant correlations between different genera's sequence abundances using the SparCC method[66], which accounts for compositional data ('sparccboot' function from the *SpiecEasi* package in R[67]). Only those genera with significant correlations (both positive and negative, with $p < 0.05$) by bootstrapping (100 times) were included in the networks. We created a separate network for each of the four treatments (N, P, N + P, and control), where a node represents a genus, and an edge between two genera represents a significant correlation in abundances between the pair. To calculate $p$-values for the network statistics in Fig. 3, we performed 10,000 permutations of the networks. To calculate $p$-values for number of links, mean degree, and connectance, we took the edge lists for all four of the networks, which consist of each possible pairwise combination of genera (those present in any of the four networks) and 0 if no link, and 1 if linked. We then randomly shuffled the edge weights (0 or 1) across pairwise treatments, keeping the nodes constant, and reformed the two focal networks from these shuffled weights. We calculated the metrics for each of the networks for each permutation. $P$-values were calculated as the proportion of permutations at least as extreme as the observed differences between the network metrics. Superscripts in Fig. 4 indicate treatments that were significant at $p < 0.05$.

**PCoAs on guild communities**. Using the 'cmdscale' function in the *vegan* package[68], we performed principal coordinate analyses (PCoAs) on Bray–Curtis distances of Hellinger transformed relative abundance data for each fungal guild and plot. PCoAs were plotted using *ggplot2*[69]. To detect significant differences among treatments and continents within each guild, we performed permutational analyses of variance (Permanova), using the 'adonis' function. Permanova analyses were run on both Bray-Curtis distances as well as on Raup-Crick transformed presence/absence data.

**Reporting summary**. Further information on research design is available in the Nature Research Reporting Summary linked to this article.

## Data availability

All data (plant, fungal, climate, and soil properties) required to repeat analyses are available in the Environmental Data Initiative (EDI) repository with the identifier https://doi.org/10.6073/pasta/8b3ff674e7123f08e0ae960d006c202e. Climate data was originally collected from the WorldClim database (version 1.4) is available at http://www.worldclim.org/bioclim. Supplementary Information is available for this paper.

## Code availability

R-code to reproduce SEM, co-occurrence analysis, ordinations, mediation tests, and the subsequent tables are available in the Environmental Data Initiative (EDI) repository with the identifier https://doi.org/10.6073/pasta/8b3ff674e7123f08e0ae960d006c202e.

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

## Acknowledgements

Authors are grateful to Nancy Johnson, Rich Philips, Jeff Gailus, and Mike McTee for helpful comments that improved earlier drafts of this manuscript. Authors also thank former undergraduate researchers Efemona Famati, Carter Linhardt, and Jessica Lettel-lier for their efforts measuring fungal colonization, CAA thanks George Arhonditsis for his support, and Y.L. and L.S.B. thank MPG Ranch for funding. This work was generated using data from the Nutrient Network (http://www.nutnet.org) experiment, funded at the site scale by individual researchers. Coordination and data management have been supported by funding to E.T.B. and E.W.S. from the National Science Foundation Research Coordination Network (NSF-DEB-1042132) and Long-Term Ecological Research (NSF-DEB-1234162 and NSF-DEB-1831944 to Cedar Creek LTER) programs, and the Institute on the Environment (DG-0001-13). We also thank the Minnesota Supercomputer Institute for hosting project data and the Institute on the Environment for hosting Network meetings. All authors are grateful to participating NutNet site scientists for generating the original data.

## Author contributions

Y.L. conceived of the idea, coordinated the analyses and wrote the first draft, C.A.A. ran the SEM and the mediation test with input from J.A.H. J.A.H. collected root colonization data, P.G.K. compiled the FUNGuild analyses, and A.D.L. conducted the co-occurrence analyses. L.S.B. plotted the PCoA and ran the PERMANOVA, and J.W.L. and N.F. provided the original sequence data. All co-authors contributed to the editing of the manuscript.

## Competing interests

The authors declare no competing interests.
