## [Peer Review File · Nature Communications]

Reviewer comments, first round –

Reviewer #1 (Remarks to the Author):

The authors present a very interesting study in which they consider 25 grasslands over the globe with plots of no nutrient addition to nitrogen and phosphorous addition. They studied the effects on fungal guilds, because fungi fulfill some crucial functions in the ecosystem. They found that nutrient addition generally promoted pathogens, reduced mutualists and did not affect saprotrophs. On top of that they found effects on the co-occurrence network within and between the fungal guilds. It further seems that the pathogens are driven by soil properties and mutualist by plant community characteristics. The authors conclude that these guild-level responses enhance our ability to predict soil functional responses to anthropogenic eutrophication and the associated longer-term responses of plant communities to this important global change factor.

I like the global approach, but it is hard to disentangle the local effects of plant communities to the general effect of the fungal guilds, and it may be that some underlying mechanisms remain uncovered. In general I find the figures convincing and clear. But my main worries are the robustness of the pattern. So, it is great that there are samples taken all over the world, but this imposes a large (site) continent effect, which does not seem to hold when assessing presence/absence data in the ordination figure 4. It also does not surprise me that the continent effects were much higher than the treatment effects within continent. Figure 1 is therefore very convincing that despite the continental effects, the treatment effects still remain on the guild level.

I have a few comments I would like to see clarified, but my general opinion is that this is a interesting thorough study with nice results that are of wider interest to the scientific community and would therefore justify a platform like Nature Communications.

Major comments

Although the flow of the story reads quite natural, I am wondering if the manuscript would benefit from a clear short sum-up of testable hypotheses. The longer versions as to why these predictions are made can then remain in the supplements. The manuscript moves now very quickly without stating an explicit question or hypotheses into a what has been done and what was found.

Could it be that the pathogen effect was in fact also plant community mediated. In plant communities with disturbed soils (grasslands in early succession or croplands) and plenty of nutrients often support pioneer plant communities that are known for their negative plant-soil feedback usually associated with plant pathogens. So, this finding where higher P is related with more pathogenic fungi does not surprise me.

While it is indicated in the ms in L60: "While direct comparisons among guilds should be made with caution due to potential amplification bias", you still do it. You indicate you "ran a separate mediation test restricted to unfertilized plots", which serves as a reference. And indeed this shows quite some significant differences that cannot be assigned to PCR-bias. But I cannot find back that you included sequencing samples including a mock community of fungi with known concentrations, to be able to indicate which groups are over- or under represented due to your PCR bias. I know that with general fungi and AMF in specific qPCRs on quantitative data have their own biases and so do PLFA's, but still I would like to see a bit more confirmation of robustness for the quantitative use of qualitative data.

It seems that the co-occurrence networks were build with 5 entries per network. Or is is replicated on the basis of the sites 25? Usually you need around 10 entries to make your network deviate from random. But if you include 25 sites per network this chance will be high. Did you test against random network creations? You mentioned you "randomly shuffled the edge weights (0 or 1) across pairwise treatments, keeping the nodes constant, and reformed the 2 focal networks from

these shuffled weights." In L260-262. I cannot fully judge if this is in fact creating two random networks or not.

Figure 4 "Results did not differ when dissimilarities were assessed based on presence/absence."

Minor comments

Why do the authors in the first paragraph do not say that eutrophication generally stimulates r-strategist plant species which pushes K-strategist plants slowly to the brink of extinction, rather than explaining that soil organisms and especially AMF will limit plant growth. The second is also the case, but the first is more of a direct eminent threat relating to species extinctions. The second, can then be used to link to the second paragraph.

There is not anything wrong with rarefying in itself, but I wonder if you also explored different options (because you loose a lot of information). There are some good alternatives out there in the literature. If you find with other methods a similar outcome, then I would trust this is a robust method.

Reviewer #2 (Remarks to the Author):

This manuscript is largely a re-analysis of published soil fungal dataset from the NutNet (Leff et al. 2015, PNAS). The new analysis showed distinct responses of the three major guilds of soil fungi (AMF, pathogens, saprotrophs) to N and P addition at the 25 grassland sites of the NutNet. Moreover, SEM analyses (at 15 sites only) showed that the responses of different fungal guilds to nutrient additions were likely caused by different factors (plant communities, soil properties, and climate). Overall, these findings improve our understanding of the responses of soil fungal community to nutrient enrichment in grassland ecosystems.

The strength of this work is that the data are from 25 sites of coordinated distributed experiments which use the same method to conduct the experiment and collect the data. In this regard, it is better than meta-analysis which uses data from different experiments with different methods and designs that may bring uncertainty to the mean results.

The main caveats of this work are 1) the majority of data were previously published (Leff et al. 2015, PNAS), and 2) the main results are largely confirmatory of a number of site-level case studies, regional-level networks (< 10 sites), or recent global meta-analyses. I do not need to list these studies as some (but not all) are already cited in the manuscript.

Specific comments

L50: Can you be more clear about the "N responses"?

L56: no data on "ecosystem function" were reported in this work.

L58: data from the first 1-4 years of treatment may only show the "short-term" response to the perturbation of nutrient addition. Longer duration of treatment would be better to tell the responses of soil communities.

L74-76: the data were generated a long time ago using the now "old" technique. If we do it today, the sequencing technology would be better and such limitation would be much lower.

L117-118: same comment as above.

L148-149: good point.

L199-200: is this number (60%) typical for similar analyses using the FUNGuild database?

Figure 1: are the root AMF data not published previously? It would be interesting to know what factors could explain the variations in the effect size among sites. Simple biplots may provide direct information on this.

Reviewer #3 (Remarks to the Author):

Summary

Utilizing a global network of grassland fertilization experiments (NutNet), this study seeks to discover the response of fertilization on soil fungal guilds in context of climate, plant community composition & root biomass, and soil properties. Structural equation modeling was used to determine the direct, and indirect, relationships among variables to determine causality in addition to other multivariate statistical tools like co-occurrence network analysis and principal coordinate analysis. The primary, novel result is fertilization increased fungal pathogens, modest suppression of soil mutualist (AMF), but had no biological impact on root AMF or saprotrophs. The primary novel conclusion, in my opinion, was the results suggest that one of the potential explanation of suppressed plant growth due to long-term fertilization is due to increased pathogen loads. Likewise, the plant-AMF relationship under fertilization may be switching from nutrient acquisition to protection from pathogens.

General Comments

The title gives the impression of a natural, observational study on eutrophication, not the precise fertilization experiments used for this study. Eutrophication suggest more than just N and P, but an excess of all nutrients. Saying worldwide I guess is okay, but the experiment is missing South American, and all Asia with a heavy focus on North America. Maybe just say "Nitrogen and phosphorus fertilization consistently favors pathogenic over mutualistic fungi in NutNet grasslands" Less catchy, but accurate.

My primary concern is the interpretation of the SEM results. While the paper strongly promotes that it is caused by shifts in plant communities, the model appears inconclusive. Pre-treatment soil properties seems to be as important as Jaccard dissimilarity or root biomass. The SEM model seems to force any fungal response through the plant response. I'm surprised that there wasn't a direct fertilization response to fungal communities. The writing in the supplementary material wasn't clear on this. Was the initial assumption that any change to the fungal community must be due to a plant response? If so, try it with a direct response from the fertilization and if this was done, make it more clear in the text using plain language. Overall I think this is an important distinction. Are drastic changes in nutrient availability a significant driver of plant-soil interaction or do primary state factors (parent material & climate) first have to be considered before considering nutrient effects?

Line

9 This is unclear and nuanced. The pathogen to AMF ratio appears caused by pathogens and not really by AMF. Are you suggest that increased pathogen loads are the cause or is it something else? If so, please provide direct supporting references on fertilization and plant disease.

107 I doubt it is pH directly, but what pH represents in regards to overall soil properties from the general type of parent material, soil redox reactions, cation exchange reactions, nutrient solubility, etc. I think this concept should be raised that it's the impact, or indicator, of pH on soil resources is likely the explanation.

118 Unclear on what is meant by "preferentially occupy roots"? Obligate AMF? Be more clear.

120 While I find this exciting, I'm unconvinced the results support this statement (AMF switching function) and comes off as speculation. More justification from empirical evidence is needed to

support this statement. Also, this assuming more soil pathogens equals more plant disease. Are these fungal plant pathogens obligate of a plant host? Can you separate pathogenic fungi into obligate and facultative pathogens? Perhaps improving soil fertility allows an abundance of facultative pathogens to flourish?

127 While there is a vague mention of habitat filtering, it is important to note that adding fertilizer would effectively homogenize the soil nutrient environment and likely reduce niche (e.g. high/low nutrient availability), thus lowering biodiversity similar to plowing and breaking up aggregates. Yes, I agree it is a disturbance by fundamentally altering the chemical spatial heterogeneity of the soil environment.

161 Again this ratio difference is promoted, but the driver appears to be just from the pathogens. If significant from both, then make a short statement that both variable are driving this response.

Figure 1 Please use a boxplot so the data is more transparent.

Revision notes

Reviewer 1.

Comment: I like the global approach, but it is hard to disentangle the local effects of plant communities to the general effect of the fungal guilds, and it may be that some underlying mechanisms remain uncovered. In general I find the figures convincing and clear. But my main worries are the robustness of the pattern. So, it is great that there are samples taken all over the world, but this imposes a large (site) continent effect, which does not seem to hold when assessing presence/absence data in the ordination figure 4. It also does not surprise me that the continent effects were much higher than the treatment effects within continent. Figure 1 is therefore very convincing that despite the continental effects, the treatment effects still remain on the guild level. I have a few comments I would like to see clarified, but my general opinion is that this is an interesting thorough study with nice results that are of wider interest to the scientific community and would therefore justify a platform like Nature Communications.

Response: We thank the Reviewer for the nice comments and summary of the results. We agree that the consistent responses to N and P addition across sites that differ greatly in plant composition, fungal communities, climate and soil abiotic conditions is a strength in this study and attest to the directional and predictable responses to fertilization in natural grasslands.

Comment: Although the flow of the story reads quite natural, I am wondering if the manuscript would benefit from a clear short sum-up of testable hypotheses. The longer versions as to why these predictions are made can then remain in the supplements. The manuscript moves now very quickly without stating an explicit question or hypotheses into a what has been done and what was found.

Response: We agree and have more or less re-written the manuscript to fully comply with NCC format requirements. Specifically, we now outline predictions and have separated different aspects of this manuscript into subheadings (Introduction, Guild responses to N and P additions, Direct or indirect effects of N and P on fungal guilds, Global drivers of fungal guilds, Limitations, Summary) to improve clarity.

Comment: Could it be that the pathogen effect was in fact also plant community mediated. In plant communities with disturbed soils (grasslands in early succession or croplands) and plenty of nutrients often support pioneer plant communities that are known for their negative plant-soil feedback usually associated with plant pathogens. So, this finding where higher P is related with more pathogenic fungi does not surprise me.

Response: Yes indeed, and our SEM indicate that the individual and combined effect of P was mediated through a change in plant community; the more N and P altered the plant community relative to control communities (i.e. greater Jaccard dissimilarity), the more pathogens were observed. Previous studies using data from many of the same plots have shown that N and P applications result in reduced root/shoot ratio (Cleland et al. *Ecosystems* 2019) and greater nutrient concentrations in the leaves and higher SLA (Firn et al. *NEE* 2019). These traits have been associated with more ruderal traits (Adler et al. *PNAS* 2014) and those plants may also be more susceptible to pathogens (Kardol et al. *Ecol Lett* 2008). We now more clearly discuss this link in lines 173-179. However, we need to point out that, for pathogens, we cannot rule out a direct effect of N and P because a model replacing plant community similarity with treatment effect performed similarly (Table S2a). This is highlighted in lines 186-188.

Comment: While it is indicated in the ms in L60: "While direct comparisons among guilds should be made with caution due to potential amplification bias", you still do it. You indicate you "ran a separate mediation test restricted to unfertilized plots", which serves as a reference. And indeed this shows quite some significant differences that cannot be assigned to PCR-bias. But I cannot find back that you

included sequencing samples including a mock community of fungi with known concentrations, to be able to indicate which groups are over- or under represented due to your PCR bias. I know that with general fungi and AMF in specific qPCRs on quantitative data have their own biases and so do PLFA's, but still I would like to see a bit more confirmation of robustness for the quantitative use of qualitative data.

Response: While common, we agree that the use of sequence counts to indirectly address relative abundance can be problematic and should be done with great care as there is not necessarily a direct correlation between sequence counts and biomass or cell numbers. The primer pair used here (ITS1f-ITS2) is arguably the most commonly used primer pair for cultivation-independent fungal community analyses, but there may be biases against fungi with large introns found in diverse members of the Ascomycota (Taylor *et al.*, 2016). The Reviewer is correct and no mock community was used in this study to assess the degree of PCR and sequencing bias, but we note that the potential biases associated with this primer pair and other primer pairs are detailed in (Bellemain *et al.*, 2010). Given the potential importance of primer biases, we returned to our FUNGuild output and examined the relative ratio of Ascomycota and Basidiomycota in our putative pathogen and saprotroph groups. We found that the Ascomycota were more abundant than the Basidiomycota in both groups at a ratio of 2.6 (saprotrophs) and 18.1 (pathogens). Because of this, and because AMF may be discriminated against when using ITS primers (Lekberg *et al.*, 2018), we have decided to remove any direct comparisons of guild abundances in our manuscript as we agree that it can be misleading. As such, we no longer report that saprotrophs are more abundant than pathogens for example. For the remainder of the analyses (responses to fertilizer treatments and climate, soil edaphic properties and plant community characteristics), however, we argue that potential PCR and sequencing bias is not an issue as the same methods are consistently applied across all samples and should therefore not influence our SEM and mediation analyses (lines 243-246). Lastly, we want to clarify one potential misunderstanding regarding the mediation test in that it cannot be used as a reference, but is simply a way to assess if and how fungal guilds change with plant community characteristics, soil edaphic properties and climatic conditions across sites when focusing only on control plots.

Comment: It seems that the co-occurrence networks were build with 5 entries per network. Or is replicated on the basis of the sites 25? Usually you need around 10 entries to make your network deviate from random. But if you include 25 sites per network this chance will be high. Did you test against random network creations? You mentioned you “randomly shuffled the edge weights (0 or 1) across pairwise treatments, keeping the nodes constant, and reformed the 2 focal networks from these shuffled weights.” In L260-262. I cannot fully judge if this is in fact creating two random networks or not.

Response: We are unsure what the reviewer means by ‘entries’. We created 4 networks- one for each treatment. Each treatment was performed at all 25 sites. Therefore, each network is made up of the significant correlations observed between genera in the plots of that treatment at all of the 25 sites. For example, the control network had genus data from control plots at all 25 sites. A node (circle) in the network represents a genus, and an edge (line) between 2 nodes represents a significant correlation in occurrence between those 2 genera. A genus was only included in the network if it was correlated to at least one other genus. The number of nodes in each network was >30, which is enough to allow deviations from random.

To determine if the networks were significantly different from each other, we compared them to random graphs. However, the way we created those random graphs ensured that each network maintained the same number of nodes, and the total number of edges across the 2 compared networks were the same. For example, to determine if the difference in edges is significant between the N network (65 edges) and the Control Network (99 edges), we need to account for the fact that the Control network had more nodes (37) than the N network (33), and thus had a greater potential for

more edges. Each pairwise combination of nodes had an edge weight (either a 1 if a connection between those 2 genera, or 0 if no connection). We reshuffled the edge weights (0's and 1's) across all genus pairs of both networks. For example, the total number of edges across both the control and N networks was 164 edges (65 + 99), and these edges were shuffled across both networks. The new random networks could now have differing numbers of edges, but the number of nodes per network remained the same, and the sum of the number of edges across both networks remained the same. We performed this random reshuffling 10,000 times, and observed the proportion of times the difference between the networks was as large as the observed difference. For example, the difference in edges between the control and N networks was $99 - 65 = 34$. We determined how many times out of the 10,000 randomly reshuffled networks the difference between these two networks was at least 34. We found that the randomly reshuffled networks had a difference of 34 or greater about 12% of the time, therefore $p=0.12$, and the difference was not significant. However, the difference in edges between the reshuffled NP and control networks was only as large as the observed difference 0.3% of the time, and therefore $p=0.003$, and was significant. R Code to repeat these analyses are provided as separate files.

Comment: Figure 4 “Results did not differ when dissimilarities were assessed based on presence/absence.”

Response: It is somewhat unclear what the issue raised here is, but we assume it is related to not clearly specifying that the Bray-Curtis analysis was based on relative abundance data. We've tried to clarify this in the figure legend now by stating “Principal coordinate analyses of Bray-Curtis distances of Hellinger transformed relative abundance data showing dissimilarities among communities for mutualists.....” and “Results were not different when dissimilarities were assessed based on Raup-Crick transformed presence/absence data.”

Comment: Why do the authors in the first paragraph do not say that eutrophication generally stimulates r-strategist plant species which pushes K-strategist plants slowly to the brink of extinction, rather than explaining that soil organisms and especially AMF will limit plant growth. The second is also the case, but the first is more of a direct eminent threat relating to species extinctions. The second, can then be used to link to the second paragraph.

Response: That is a good point and we have added that nutrients can cause shift in plant communities toward ruderal plants (lines 55-58). We also now discuss the link between ruderal plants and shifts in fungal guilds more explicitly, and show that there is support for a shift toward plants with more ruderal traits in NutNet plots receiving N and P, which can help explain the shift in pathogen/mutualist ratios based on susceptibility to disease and AM dependency (lines 173-179). We have opted to keep the suggestion of a potential role of soil microbes at the end of the first paragraph as this is one of the key topics of the paper, which focused on belowground, not aboveground, responses to fertilization.

Comment: There is not anything wrong with rarefying in itself, but I wonder if you also explored different options (because you lose a lot of information). There are some good alternatives out there in the literature. If you find with other methods a similar outcome, then I would trust this is a robust method.

Response: Indeed, there are many different ways to deal with unequal sequence counts across samples. We did not explore other normalization methods for two reasons. First, while losing sensitivity, rarefaction is a consistent, simple, and commonly used method that has been shown to have lower false discovery rates relative to other normalization methods (Weiss *et al.*, 2017). Because we observed significant treatment effects, one can argue that they are more robust because of the loss in sensitivity and information that randomization involves. Second, we wanted to keep our analyses as similar as possible to the original publication by Leff *et al.* (2015) that relied on rarefactions to allow for greater

comparisons. The comment by the Reviewer prompted us to construct rarefaction curves to assess to what degree the fungal communities were characterized across all the samples (see below). While we did not characterize all fungal taxa in all samples (an impossible task regardless of the methods employed), we are confident that we identified the more abundant OTUs across sites. Because we believe this is relevant information for readers to judge the robustness of our results, we have added this figure to the supplemental material (Fig. S6).

Figure 1. Rarefaction curves of all samples collected across all sites and treatments. It suggests that, while not all taxa may have been identified at all sites at the chosen 485 reads per sample rarefaction cut-off, the abundant taxa were most likely included.

Reviewer 2

Comment: This manuscript is largely a re-analysis of published soil fungal dataset from the NutNet (Leff et al. 2015, PNAS). The new analysis showed distinct responses of the three major guilds of soil fungi (AMF, pathogens, saprotrophs) to N and P addition at the 25 grassland sites of the NutNet. Moreover, SEM analyses (at 15 sites only) showed that the responses of different fungal guilds to nutrient additions were likely caused by different factors (plant communities, soil properties, and climate). Overall, these findings improve our understanding of the responses of soil fungal community to nutrient enrichment in grassland ecosystems. The strength of this work is that the data are from 25 sites of coordinated distributed experiments which use the same method to conduct the experiment and collect the data. In this regard, it is better than meta-analysis which uses data from different experiments with different methods and designs that may bring uncertainty to the mean results.

Response: Thank you for the encouraging evaluation of our paper. We want to clarify a possible misunderstanding with the SEM, however, as this analysis showed that guild responses to N and P were primarily mediated by shifts in plants rather than direct effects of nutrient addition (although for pathogens, the model including direct effects performed equally well, now discussed in lines 186-188). The effects of plant communities, soil properties, and climate that the Reviewer is referring to were assessed with our mediation test, which attempted to identify potential underlying drivers for differences in guild abundances *across* sites when analyses were restricted to unfertilized plots only.

Comment: The main caveats of this work are 1) the majority of data were previously published (Leff et al. 2015, PNAS), and 2) the main results are largely confirmatory of a number of site-level case studies, regional-level networks (< 10 sites), or recent global meta-analyses. I do not need to list these studies as some (but not all) are already cited in the manuscript.

Response: We agree that there is some overlap between Leff et al. 2015 and this publication. However, we believe that we add significantly to that previous publication in several important ways. First, while the Leff publication showed different responses among fungal phyla, only the analyses focused on the phylum Glomeromycota was informative about the potential function of these higher level taxa. As such, the original paper has limited information about shifts in the potential functional attributes of the fungal communities, which we show here. Second, the underlying drivers, i.e. whether ‘shifts in plant community composition drive shifts in fungal communities or both respond similarly to changes in edaphic factors’ remains an open question in Leff et al. We address this via the SEM. Third, we include novel data involving AMF root colonization, which allowed us to quantify potential shifts in fungal biomass allocation between roots and soil. And fourth, our mediation test adds information about potential drivers of fungal guild distributions across grasslands that differ in plant communities, abiotic conditions and climate, with our network analyses suggesting that potential fungal interactions are disrupted (at least temporarily) by nutrient addition. As such, we believe that our manuscript adds important novel information to justify publication. Nonetheless, acknowledging that there is some overlap, we have edited the section where we first introduce the Leff paper to be more forthcoming about that paper and to outline what specifically we re-analyzed and how, what data was novel, and how this paper build on the Leff et al. paper (lines 91-113). The literature related to PSF have only recently started to assess the context-dependency of these feedbacks and we are encouraged that our data largely support previous studies, based either on meta-analyses, single sites, or multiple sites across ecosystem types.

Comment: Can you be more clear about the “N responses”?

Response: We now outline that “N additions can affect saprotrophic activity and decomposition rates” (lines 80-81).

Comment: No data on “ecosystem function” were reported in this work.

Response: That is a good point and the sentence is now re-written as: “To better predict current and future functioning of soil microbial communities, we need to understand how altered nutrient availability influences fungal guilds, if responses are consistent across locations, and the nature of the underlying drivers” (lines 88-91).

Comment: Data from the first 1-4 years of treatment may only show the “short-term” response to the perturbation of nutrient addition. Longer duration of treatment would be better to tell the responses of soil communities.

Response: That is true. To address this potential temporal issue within the limitations of this study, we tested whether sites where N and P had been applied longer showed stronger responses. This test showed that this was not the case for any of the guilds ($P > 0.05$, now discussed in lines 159-162). We have added information on these additional analyses in lines 163-165 of the SI, and we also make sure to point out that the observed responses are relatively short-term (lines 238-240).

Comment: the data were generated a long time ago using the now “old” technique. If we do it today, the sequencing technology would be better and such limitation would be much lower.

Response: Indeed. We argue, however, that there is still valuable information that can be gained from this dataset, but unfortunately this does not include careful community analyses within each guild. It is our hope that future samplings of NutNet sites will allow for this as that data would allow us to answer important questions about shifts within guilds as well as short- versus long-term responses of fungal communities the Reviewer highlighted above.

Comment: Is this number (60%) typical for similar analyses using the FUNGuild database?

Response: This is indeed very typical for the amount of fungal OTUs that can be assigned to guilds using FUNGuild. In fact, in the original FUNGuild paper (Nguyen *et al.*, 2016) assignment percentages ranged from 40-60% depending on habitat for the three datasets compared (with the grassland habitat being only 40%). As such, the 60% assignment in this dataset is relatively good and we feel confident the results obtained are robust.

Comment: Figure 1: are the root AMF data not published previously? It would be interesting to know what factors could explain the variations in the effect size among sites. Simple biplots may provide direct information on this.

Response: The AMF root colonization data has not been published elsewhere so we agree that these data deserve more attention. We now include two supplemental figures. The first (Fig. S4 with plotted results from Table S5) highlights the significant relationships that we found in our global analysis looking at only the control plots, including biplots between AMF root colonization and significant soil properties (soil iron content) and plant community metrics (aboveground legume biomass and total root biomass). The second plot (Fig. S5) shows differences in AMF root colonization across sites.

Reviewer 3

Comment: The title gives the impression of a natural, observational study on eutrophication, not the precise fertilization experiments used for this study. Eutrophication suggest more than just N and P, but an excess of all nutrients. Saying worldwide I guess is okay, but the experiment is missing South American, and all Asia with a heavy focus on North America. Maybe just say “Nitrogen and phosphorus fertilization consistently favors pathogenic over mutualistic fungi in NutNet grasslands” Less catchy, but accurate.

Response: We agree and have changed the title to “Nitrogen and phosphorus fertilization consistently favor pathogenic over mutualistic fungi in grassland soils”.

Comment: My primary concern is the interpretation of the SEM results. While the paper strongly promotes that it is caused by shifts in plant communities, the model appears inconclusive. Pre-treatment soil properties seems to be as important as Jaccard dissimilarity or root biomass. The SEM model seems to force any fungal response through the plant response. I'm surprised that there wasn't a direct fertilization response to fungal communities. The writing in the supplementary material wasn't clear on this. Was the initial assumption that any change to the fungal community must be due to a plant response? If so, try it with a direct response from the fertilization and if this was done, make it more clear in the text using plain language. Overall I think this is an important distinction. Are drastic changes in nutrient availability a significant driver of plant-soil interaction or do primary state factors (parent material & climate) first have to be considered before considering nutrient effects?

Response: Based on our literature review, our initial hypothesis was that changes in fungal guilds were primarily driven by plant communities. However, we also systematically tested alternative hypotheses. Following the reviewer's suggestion, we revisited our models and compared the amount of variance explained by alternative models presented in Table S2a. As suggested by the reviewer, we found a smaller, but similar, amount of variability explained when replacing plant descriptors with treatment. We adjust our argument accordingly (by adding the word “often” in lines 39, and again in lines 186-188) and also highlight the fact that separating cause from effect would require detailed time-series data (lines 183-186). We reiterate, however, that consistent responses by pathogens and AMF soil colonization were found *despite* large differences in plant and fungal communities as well as large differences in climate and pre-treatment soil conditions.

Comment: I doubt it is pH directly, but what pH represents in regards to overall soil properties from the general type of parent material, soil redox reactions, cation exchange reactions, nutrient solubility, etc. I think this concept should be raised that it's the impact, or indicator, of pH on soil resources is likely the explanation.

Response: We agree and have changed this to read “Why pH—or other soil properties that correlate with soil pH—affects one guild more than another.... ” (lines 203-206).

Comment: Unclear on what is meant by “preferentially occupy roots”? Obligate AMF? Be more clear

Response: Yes, sorry about that confusion. We have added a sentence stating: “To acquire C, AMF must colonize roots, but relative biomass allocation between roots and soil differs among fungal taxa and environmental conditions”. (lines 133-135).

Comment: While I find this exciting, I'm unconvinced the results support this statement (AMF switching function) and comes off as speculation. More justification from empirical evidence is needed to support this statement.

Response: It *is* speculation at this point and we have altered the sentence to read: “Thus, we speculate that an intriguing alternative hypothesis to nutrient-induced parasitism is that N promotes pathogens....” (lines 145-147). We have decided to keep this in for two main reasons; 1) it is consistent with shifts in community composition observed in several studies with N addition and what we know about biomass allocation and function of these taxa, 2) our results could have important implications for the 'parasitism to mutualism' continuum idea and therefore the idea is worth putting out there to either be supported or refuted with future work that builds on our study.

Comment: Also, this assuming more soil pathogens equals more plant disease. Are these fungal plant pathogens obligate of a plant host? Can you separate pathogenic fungi into obligate and facultative pathogens? Perhaps improving soil fertility allows an abundance of facultative pathogens to flourish?

Response: Unfortunately parsing obligate versus facultative pathogens is beyond the scope of what is possible with our analyses given the current limitations of the FUNGuild database. However, there are studies that have found positive associations between pathogenic fungal OTU abundance and both disease severity (Busby et al. 2016) as well as plant yield loss (Li et al. 2014, both now cited), suggesting that our presumption about increased pathogen OTU abundance being positively correlated with greater disease is likely ecologically valid. We now address this in lines 259-262.

Comment: While there is a vague mention of habitat filtering, it is important to note that adding fertilizer would effectively homogenize the soil nutrient environment and likely reduce niche (e.g. high/low nutrient availability), thus lowering biodiversity similar to plowing and breaking up aggregates. Yes, I agree it is a disturbance by fundamentally altering the chemical spatial heterogeneity of the soil environment.

Response: The potential for reduced niches and a more homogeneous environment with fertilizer addition are good points, and we now outline this a bit more: “Network complexity can be reduced by perturbation and fungal communities may respond to fertilization as a disturbance (possibly short-term), where the enriched soil nutrient environment is completely altered and homogenized in ways that could alter the available niches. Our results could be a product of habitat filtering where taxa respond similarly to environmental shifts, but do not necessarily interact.” (lines 208-212).

Comment: Again this ratio difference is promoted, but the driver appears to be just from the pathogens. If significant from both, then make a short statement that both variables are driving this response.

Response: We now outline that the response is driven by both pathogens and mutualists but that the response by mutualists is smaller: “Direct links between fungal guilds and plant soil feedback have been shown previously. The promotion by pathogens and—to a lesser extent—the suppression of AMF with nutrient addition observed here may help explain why plant biomass responses to fertilizer decline with time. It also supports predictions that effects of soil biota on plant growth are more negative in resource-rich environments.” (lines 129-133).

Comment: Figure 1 Please use a boxplot so the data is more transparent

Response: We added boxplots of the partial residual to the figure. Thanks for the suggestion.

References

- Bellemain E, Carlsen T, Brochmann C, Coissac E, Paberlet P, Kauserud H. 2010.** ITS as an environmental DNA barcode for fungi: an in silico approach reveals potential PCR biases. *BMC Microbiology* **10**: 1–9.
- Leff JW, Jones SE, Prober SM, Barberán A, Borer ET, Firn JL, Harpole WS, Hobbie SE, Hofmockel KS, Knops JMH, et al. 2015.** Consistent responses of soil microbial communities to elevated nutrient inputs in grasslands across the globe. *Proceedings of the National Academy of Sciences* **112**: 10967–10972.
- Lekberg Y, Vasar M, Bullington LS, Sepp SK, Antunes PM, Bunn R, Larkin BG, Öpik M. 2018.** More bang for the buck? Can arbuscular mycorrhizal fungal communities be characterized adequately alongside other fungi using general fungal primers? *New Phytologist* **220**: 971–976.
- Nguyen NH, Song Z, Bates ST, Branco S, Tedersoo L, Menke J, Schilling JS, Kennedy PG. 2016.** FUNGuild: An open annotation tool for parsing fungal community datasets by ecological guild. *Fungal Ecology* **20**: 241–248.
- Taylor DL, Walters WA, Lennon NJ, Bochicchio J, Krohn A, Caporaso JG, Pennanen T. 2016.** Accurate Estimation of Fungal Diversity and Abundance through Improved Lineage-Specific Primers Optimized for Illumina Amplicon Sequencing (D Cullen, Ed.). *Applied and Environmental Microbiology* **82**: 7217–7226.
- Weiss S, Xu ZZ, Peddada S, Amir A, Bittinger K, Gonzalez A, Lozupone C, Zaneveld JR, Vázquez-Baeza Y, Birmingham A, et al. 2017.** Normalization and microbial differential abundance strategies depend upon data characteristics. *Microbiome* **5**: 27.

Reviewer comments, second round –

Reviewer #1 (Remarks to the Author):

Dear authors,

I thoroughly went through all your responses to my raised comments and traced them back in the text of the manuscript. I am very happy that all of my comments were taken seriously, and most also led to improvement of the clarity of the manuscript. I could also not recall anymore what my point was about figure 4, probably some unfinished thought process from my side. I am also convinced now on the robustness of the networks after your explanation.

For me all my comments are addressed and I have no further comments. I think the manuscript is fit for publication.

Elly Morriën

Reviewer #2 (Remarks to the Author):

The authors did a good job of answering my comments and revising the manuscript. Particularly, they better explained the differences of this study with an earlier paper (Leff et al. 2015 PNAS) and clarified the new findings of this study. They also clearly described the SEM and the mediation test. I think the findings are novel and could inform future studies on soil fungal communities in response to global changes. I would like to recommend a recent meta-analysis paper (Han YF, Feng JG, Han MG, Zhu B. 2020. Responses of arbuscular mycorrhizal fungi to nitrogen addition: a meta-analysis. *Global Change Biology* 26:7229-7241) on the responses of AMF to N addition at global scale (including grasslands) that may be worthwhile to cite. Overall, I suggest publication of this work.

Reviewer #3 (Remarks to the Author):

I am satisfied with the author's response to the reviewers comments and corrections made to the manuscript. I have no significant concerns.